# Abnormal spirometric patterns and respiratory symptoms in HIV patients with no recent pulmonary infection in a periurban hospital in Ghana

**Kwame Yeboah** [1]*, **Latif Musa**[1,2], **Kweku Bedu-Addo**[2]

**1** Department of Physiology, University of Ghana Medical School, Accra, Ghana, **2** Department of Physiology, School of Medicine and Dentistry, KNUST, Kumasi, Ghana

* kyeboah@ug.edu.gh, melvinky@gmail.com

## Abstract

### Background

Human immunodeficiency virus (HIV) infection is associated with chronic airway obstruction, even in patients who have achieved viral suppression from combination antiretroviral treatment (cART). Spirometry is a supplementary test that aids in diagnosing pulmonary dysfunction in people living with HIV.

### Aim

To compare the prevalence of spirometric abnormalities among cART-treated HIV patients and cART-naïve HIV patients with non-HIV controls with no recent history of pulmonary infection in a peri-urban hospital in Ghana.

### Methods

In a case-control design, spirometry was performed in 158 cART-treated HIV patients, 150 cART-naïve HIV patients and 156 non-HIV controls. Clinical, sociodemographic data and respiratory symptoms were collected using a structured questionnaire. Spirometric abnormalities were categorised as obstructive (OSP) or restrictive (RSP) spirometric patterns based on the Cameroonian reference equation.

### Results

The prevalence of OSP was higher in the cART-treated and cART-naïve HIV patients compared to non-HIV controls (13.9% vs 10.7% vs 5.1% respectively, p = 0.026), whereas that of RSP was similar among the study groups. Respiratory symptoms were common among cART-treated and cART-naïve HIV patients compared to non-HIV controls (48.1% vs 40% vs 19.2% respectively, p < 0.001). The major factors associated with OSP were female gender [OR (95% CI) = 2.46 (1.09–5.13), p = 0.031], former cigarette smoking [1.92 (1.04–3.89), p < 0.001], exposure to medium-to-high levels of biomass [3.07 (1.16–8.73), p = 0.019], presence of a respiratory symptom [1.89 (1.11–5.08), p = 0.029] and unemployment

**Data Availability Statement:** The dataset supporting the conclusion of this study is available at the OSF platform with the link: https://osf.io/

bg7r2/?view_only=
f349189f097c4f00bf46c13933760a30.

**Funding:** The authors received no specific funding
for this work.

**Competing interests:** The authors have declared
that no competing interests exist.

**Abbreviations:** cART, combination antiretroviral
therapy; FEV1, forced expiratory volume in 1s; FVC,
forced vital capacity; GLI, Global Lung Initiative;
HIV, human immunodeficiency virus; LLN, lower
limit of normal; OSP, obstructive spirometric
pattern; PLWH, people living with HIV; RSP,
restrictive spirometric patterns; SSA, sub-Saharan
Africa.

[3.26 (1.19–8.95), p = 0.042]. The major determinants of RSP were age, female gender [1.74 (1.05–4.29), p = 0.041], former cigarette smoking [2.31 (1.27–6.77), p < 0.001] and medium-to-high biomass exposure [1.58 (1.06–5.37), p = 0.043].

## Conclusion

In HIV patients without any recent pulmonary infection in a peri-urban area of Ghana, there was a higher prevalence of OSP among cART-treated and cART naïve HIV patients compared to the non-HIV control. However, the prevalence of RSP was similar among HIV patients and non-HIV controls.

## Introduction

Sub-Saharan Africa (SSA) has the largest population of people living with human immuno-deficiency virus (HIV); this is associated with healthcare, socio-economic and developmental challenges [1]. It was reported in 2017 that 71% of the global total number of people living with HIV (PLWH) resides in SSA, with 75% of deaths and 65% of new HIV infections occurring in the region [2]. In Ghana, the prevalence of HIV infection is 1.7% and is concentrated mostly in urban areas [3]. With the introduction of effective combination antiretroviral therapy (cART), PLWH now lives longer due to decreased morbidity and mortality from opportunistic infections [4]. This has increased the susceptibility of PLWH to chronic diseases at greater rates than those observed in HIV-uninfected persons, making HIV infection a manageable chronic disease [5]. Chronic obstructive pulmonary disease (COPD) is one of the major causes of death and disability [2] with an estimated global prevalence of 10.3% [6]. HIV infection has been associated with an increased prevalence of COPD; previous studies suggesting that the prevalence of COPD is 21% higher in PLWH compared to the general population [7]. The high burden of COPD persists in PLWH even after adjustments for risk factors such as smoking, opportunistic infections and injection drug use [4, 8–11].

Spirometry is a simple, effective and reproducible method that can aid the diagnosis of COPD, and the results can be categorized into two major patterns of abnormalities. An obstructive spirometric pattern (OSP) is characterized by a disproportional reduction in the forced expiratory volume in 1 second (FEV1) relative to the forced vital capacity (FVC), leading to a reduction in the FEV1/FVC ratio to less than the lower limit of normal (LLN) of a similar population [12, 13]. A restrictive spirometric pattern (RSP) is defined by a normal FEV1/FVC (> LLN), but an FVC is less than 80% of the predicted value. Both OSP and RSP are associated with increased mortality in the general population [13, 14]. Despite the high prevalence of PLWH in SSA, few studies have reported the prevalence and risk factors for these abnormal spirometric patterns with recent reports from the FRESH AIR study in Uganda [5, 7, 8, 15–18]. Most of these studies attributed abnormal spirometry in HIV to opportunistic pulmonary infections. This study compared the prevalence of spirometric abnormalities among HIV patients on cART treatment and cART-naïve HIV patients with non-HIV controls with no recent history of pulmonary infection in a peri-urban hospital in Ghana. The study hypothesized that HIV infection would increase the prevalence of spirometric abnormalities compared to those with no HIV infection. Also, the study investigated various factors associated with spirometric abnormalities in PLWH in Ghana.

## Methods

### Study participants and design

The study used a case-control design, conducted from October 2019 through February 2020, with HIV patients as cases and the controls were non-HIV individuals who visited the facility for voluntary testing of their HIV status. HIV patients were categorized as those on cART management and cART-naïve HIV patients who have not been on cART treatment for more than 4 weeks. The study was conducted at Atua Government Hospital, a 150-bed primary healthcare facility, located in the Lower Manya Krobo Municipality in the Eastern region of Ghana. The Lower Manya Krobo Municipality has a higher prevalence of HIV infection (7%) compared to the national prevalence which stands at 1.7% in 2020 [3]. The hospital has about 3000 HIV patients on its register. Ethical approval was granted by the Korle Bu Teaching Hospital Institutional Review Board (Protocol ID number: KBTH-IRB/000131-2018) and all participants provided voluntary informed consent before joining the study.

### Inclusion and exclusion criteria

Participants younger than 20 years, those diagnosed with severe heart diseases or hospitalized for cardiopulmonary complications within the past year, those with a diagnosis of lung infection or disease from their clinical case notes or answering "yes" to the question: "Have you been told by your health worker that you have lung disease?" were excluded from the study. In addition, pregnant women, any individual with mental illness, active or recent (within the past 24 months) tuberculosis or lung infection and those with other conditions contraindicated for spirometry were excluded from the study. Initially, 180 cART-treated HIV patients, 170 cART-naïve controls and 170 non-HIV controls were recruited into the study. However, 22 cART-treated HIV patients, 20 cART-naïve HIV patients and 12 non-HIV controls failed to produce the appropriate manoeuvres required for accurate spirometry recordings, and hence, they were excluded from the analysis.

### Data collection

A structured questionnaire was used to obtain data on sociodemographic factors like age, gender, lifestyle factors (smoking, alcohol intake), medical history, current HIV medication, occupation, education, marital status and biomass exposure. Medium-high biomass exposure was defined as those who reported using firewood, sawdust, cow dung or corn cubs as a fuel source for work or cooking at home on most days [19]. Respiratory symptoms were assessed using a modified version of the American Thoracic Society respiratory questionnaire, which included the following questions: "Do you usually have coughs without cold for more than 2 months in a year?", "Do you usually bring up phlegm from your chest when you don't have a cold?", "Do you have difficulty breathing when you walk fast, use a staircase or run?" and "Have you ever had an attack of wheezing or whistling in your chest that made you feel short of breath?" [20]. These questions were administered in the local dialect for participants with limited formal education.

Body weight and height were measured using a stadiometer in light clothing with footwear removed, with body mass index (BMI) calculated as weight/height$^2$ and categorized as underweight ($< 18.5$ kg/m$^2$), normal weight (18.5–24.9 kg/m$^2$), overweight (25–29.9 kg/m$^2$), and obese ($\geq 30$ kg/m$^2$). Hypertension was defined as self-reported use of antihypertensive treatment and/or systolic blood pressure $\geq 140$ mmHg and/or diastolic blood pressure $\geq 90$ mmHg. Venous blood samples were collected from HIV patients to measure CD4 + lymphocytes.

Spirometry was performed by a qualified technician using a Vitalograph Pneumotrac desktop spirometer model 6800 following the ATS/ERS 2019 standard guidelines [21]. Abnormal spirometric patterns were computed based on fixed ratio and lower limits of normal (LLN) using the Cameroonian spirometric equation [22]. The normal spirometric pattern was defined by $FEV_1$, FVC, and $FEV_1$/FVC all > LLN. Obstructive spirometric pattern (OSP) was defined as $FEV_1$/FVC < 0.7 with a percentage predicted FVC > 80% using the fixed ratio or $FEV_1$/FVC < LLN with FVC > LLN using the LLN method. A restrictive spirometric pattern was defined as $FEV_1$/FVC > 0.7 with a percentage predicted FVC < 80% by fixed ratio or FVC < LLN with $FEV_1$/FVC > LLN [22].

## Sample size

The sample size required was calculated using two studies conducted in rural Uganda that reported the prevalence of OSP in HIV patients to be 6.2% [7] and that of the non-HIV population to be 16.2% [23]. Therefore, a minimum of 153 participants would be required to achieve a power of 80% at a 95% confidence level with an estimated odds ratio of 2.93.

## Data analysis

Data were analysed using IBM SPSS version 28. Continuous descriptive data were presented as means and standard deviations for normally distributed data and comparisons between groups were made using one-way ANOVA, while non-normally distributed data were presented as medians and interquartile ranges and comparisons were made using the Kruskal-Wallis' test. Categorical descriptive data were presented as frequencies and percentages and analysed using the chi-square test. Univariate and multivariable logistic regression models were used to analyse the odds of participants with various socio-demographic and clinical factors having abnormal spirometry and respiratory symptoms. A p-value < 0.05 was considered statistically significant.

## Results

### General characteristics of study participants

The age range of participants was 20–70 years with a mean age of 38.4±13.7 years. As shown in Table 1, there was no difference in mean age among various categories of participants. There was a high proportion of females (67.2%) in this study participants and participants who currently (5.2%) or formerly (15.6%) smoked were mostly HIV patients. HIV patients were mostly unemployed (11%) or self-employed (43.5%) and had no formal education (14.9%) or attended school up to junior high level (59.7%). HIV patients were mostly underweight (9.1%) and had greater biomass exposure from work or home (31.8%). Measurements from spirometry indicated that cART-treated HIV patients had reduced $FEV_1$, FVC and $FEF_{25-75}$ l/min compared to cART-naïve HIV patients and non-HIV participants. Compared to the non-HIV controls, the prevalence of OSP was high in cART-naïve and cART-treated HIV patients using the LLN. There was no difference in the prevalence of RSP using the LLN. When the fixed ratio was used to determine spirometric abnormalities, there was no difference in the prevalence of OSP and RSP among the study groups (Table 1). The average duration of HIV infection in cART-treated HIV patients was 7.6±4.6 years and the average duration of cART treatment was 7.2±4.5 years. For the cART medication regimen, 94 (59.5%) patients were treated with TDF/3TC/NVP or EFV regimens, 52 (32.9%) patients were on AZT/3TC/NVP or EFV regimens and 12 (7.6%) patients were on LPV/r-based regimens.

**Table 1. General characteristics of study participants.**

| | All Participants (n = 464) | Non-HIV participants (n = 156) | cART-naïve HIV patients (n = 150) | cART-treated HIV patients (n = 158) | p |
|---|---|---|---|---|---|
| Age, years | 38.4±13.7 | 36.7±14.4 | 38.2±11.6 | 39±11.4 | 0.109 |
| Females, n (%) | 312 (67.2) | 106 (67.9) | 84 (56) | 122 (77.2) | 0.02 |
| Smoking, n (%) | | | | | 0.029 |
| Current | 16 (3.4) | 2 (1.3) | 4 (2.7) | 10 (6.3) | |
| Former | 57 (15.9) | 9 (5.8) | 22 (14.6) | 26 (16.5) | |
| Never | 187 (80.6) | 71 (92.9) | 124 (82.7) | 132 (83.5) | |
| Alcohol intake, n (%) | 102 (22) | 38 (24.4) | 36 (24) | 28 (17.7) | 0.55 |
| Married, n (%) | 198 (42.7) | 70 (44.9) | 62 (41.3) | 66 (41.8) | 0.79 |
| Education, n (%) | | | | | <0.001 |
| None | 50 (10.8) | 4 (2.6) | 14 (9.3) | 32 (20.3) | |
| Up to JHS | 208 (44.8) | 24 (15.4) | 80 (53.3) | 104 (65.8) | |
| SHS to tertiary | 68 (14.7) | 24 (15.4) | 24 (16) | 20 (12.7) | |
| Tertiary | 138 (29.7) | 104 (66.7) | 32 (21.3) | 2 (1.3) | |
| Employment, n (%) | | | | | <0.001 |
| Unemployed | 56 (12.1) | 22 (14.1) | 20 (13.3) | 14 (8.9) | |
| Self-employed | 148 (31.9) | 14 (9) | 54 (36) | 80 (50.6) | |
| Employed | 260 (56) | 120 (76.9) | 76 (50.7) | 64 (40.5) | |
| Biomass exposure, n (%) | | | | | <0.001 |
| Low | 342 (73.7) | 132 (84.6) | 96 (64) | 114 (72.2) | |
| Medium-high | 122 (26.3) | 24 (15.4) | 54 (36) | 44 (27.8) | |
| Weight, kg | 60±13.2 | 68±13.3 | 64.4±7.3 | 65.8±14 | 0.204 |
| BMI, kg/m$^2$ | 24.8±5 | 25.4±4.7 | 23.7±4.4 | 25.3±5.7 | 0.061 |
| BMI categories, n (%) | | | | | 0.024 |
| Underweight | 34 (7.4) | 6 (3.9) | 16 (10.7) | 12 (7.6) | |
| Normal | 238 (51.5) | 70 (45.5) | 94 (62.7) | 74 (46.8) | |
| Overweight | 116 (25.1) | 46 (29.9) | 18 (12) | 52 (32.9) | |
| Obese | 74 (16) | 32 (20.8) | 22 (14.7) | 20 (12.7) | |
| Current CD4 count, cells/mm$^2$ | 405 (273–562) | | 403 (253–583) | 430 (327–534) | 0.804 |
| Spirometric measurements | | | | | |
| FEV1, l | 2.2±0.6 | 2.4±0.6 | 2.3±0.7 | 1.9±0.9*# | <0.001 |
| FVC, l | 2.7±0.7 | 3±0.7 | 2.9±0.8 | 2.3±0.5*# | <0.001 |
| FEF$_{25–75}$, l/min | 2.3±0.9 | 2.5±0.9 | 2.4±1 | 1.9±0.8*# | <0.001 |
| FEV1/FVC, % | 80.3±6.9 | 81.3±5.7 | 80±7.4 | 79.4±7.3 | 0.129 |
| Abnormal spirometry by LLN, n (%) | | | | | |
| Obstructive | 46 (9.9) | 8 (5.1) | 16 (10.7)* | 22 (13.9)* | 0.026 |
| Restrictive | 24 (19.8) | 26 (16.7) | 20 (13.3) | 36 (22.8) | 0.087 |
| Abnormal spirometry by a fixed ratio, n (%) | | | | | |
| Obstructive | 26 (5.6) | 4 (2.6) | 8 (5.3) | 14 (8.9) | 0.052 |
| Restrictive | 92 (19.8) | 28 (17.9) | 26 (17.3) | 38 (24.1) | 0.258 |
| Respiratory symptoms, n (%) | 166 (35.8) | 30 (19.2) | 60 (40) | 76 (48.1) | <0.001 |

cART, combination antiretroviral therapy; JHS, junior high school; SHS, senior high school; BMI, body mass index; FEV1, forced expiratory volume in 1 second; FVC, forced vital capacity.

*: p < 0.05 compared to non-HIV

#: p < 0.05 compared to untreated HIV patients

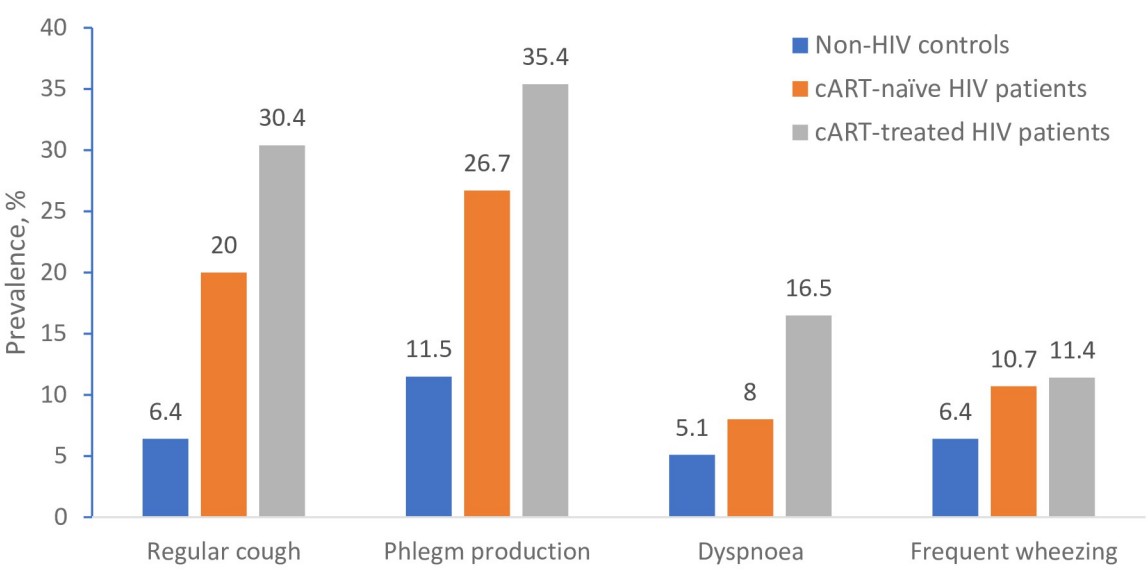

**Fig 1. Prevalence of respiratory symptoms in study participants by the HIV status.** HIV status was associated with regular cough ($\chi^2$ = 29.5, p<0.001), phlegm production ($\chi^2$ = 27.4, p<0.001) and dyspnoea ($\chi^2$ = 12.2, p = 0.002), but no association with frequent wheeze ($\chi^2$ = 4.8, p = 0.091).

## Respiratory symptoms

Among the entire study participants, 35.8% reported the presence of at least one respiratory symptom, with a higher proportion in cART-treated and cART-naïve HIV patients compared to non-HIV participants (48.1% vs 40% vs 19.2 respectively, p < 0.001). Phlegm production and regular cough were commonly reported in 26.4% and 19% of the entire study participants respectively, with a higher proportion in cART-treated and cART-naïve HIV patients compared to non-HIV participants (phlegm: 35.4% vs 26.7% vs 6.4%, p < 0.001; regular coughs: 30.4% vs 20% vs 11.5%, p < 0.001 respectively). Similar trends were observed in participants who reported having dyspnoea on exertion and frequent wheeze symptoms (Fig 1). Increasing age, being unemployed, former cigarette smoking, exposure to a high-medium level of biomass, being HIV patients with or without cART treatment and having basic or no education was associated with increased odds of having respiratory symptoms in unadjusted regression models. In the adjusted regression models, exposure to medium-high levels of biomass, being a cART-treated HIV or cART-naïve HIV patient were associated with increased odds of having respiratory symptoms (Table 2).

## Factors associated with abnormal spirometric patterns

In logistic regression analyses, female gender, unmarried, a former cigarette smoker, unemployed, having no formal education, underweight and having respiratory symptoms were associated with increased odds of having OSP in the unadjusted models. In the adjusted models, female gender, former cigarette smoking, medium-to-high biomass exposure, presence of respiratory symptoms, unemployed and being underweight were associated with increased odds of having OSP (Table 3). The associated factors of OSP among various categories of

**Table 2. Factors associated with respiratory symptoms from logistic regression analyses.**

| | Unadjusted model | | Adjusted model | |
|---|---|---|---|---|
| | OR (95% CI) | p | OR (95% CI) | p |
| Age (per 1 yr increase) | 1.02 (1.01–1.03) | 0.007 | 1 (0.98–1.02) | 0.734 |
| Females (reference = males) | 1.32 (0.87–1.99) | 0.189 | 1.14 (0.69–1.88) | 0.623 |
| Unmarried (reference = married) | 1.34 (0.91–1.97) | 0.14 | 1.18 (0.76–1.86) | 0.462 |
| Alcohol intake (reference = non-drinkers) | 0.97 (0.62–1.54) | 0.909 | 1 (0.59–1.68) | 0.986 |
| Smoking (reference: never smoked) | | | | |
| Current | 1.84 (0.68–4.98) | 0.233 | 1.52 (0.5–4.64) | 0.463 |
| Former | 2.75 (1.12–5.08) | <0.001 | 1.94 (1–4.65) | 0.071 |
| Medium/high biomass exposure (reference = low biomass exposure) | 2.72 (1.12–4.57) | 0.008 | 1.87 (1.07–3.49) | 0.037 |
| HIV status (reference = non-HIV) | | | | |
| cART-naïve | 2.8 (1.67–4.69) | <0.001 | 2.03 (1.1–3.74) | 0.023 |
| cART-treated | 3.89 (2.35–6.46) | <0.001 | 2.61 (1.32–5.17) | 0.006 |
| Employment (reference = Employed) | | | | |
| Self-employed | 1.4 (0.92–2.13) | 0.118 | 0.83 (0.5–1.36) | 0.451 |
| Unemployed | 1.88 (1.05–3.38) | 0.035 | 1.78 (0.91–3.48) | 0.091 |
| Education (reference = tertiary) | | | | |
| None | 3.32 (1.67–6.6) | <0.001 | 2.22 (0.89–5.51) | 0.086 |
| Basic/JHS | 3.21 (1.97–5.22) | <0.001 | 1.79 (0.89–3.62) | 0.102 |
| SHS/Tech | 0.93 (0.46–1.91) | 0.85 | 0.62 (0.27–1.43) | 0.263 |
| BMI categories (reference: normal) | | | | |
| Underweight | 1.85 (0.9–3.81) | 0.095 | 1.58 (0.73–3.42) | 0.25 |
| Overweight | 0.68 (0.42–1.1) | 0.116 | 0.6 (0.35–1.03) | 0.062 |
| Obese | 0.79 (0.45–1.37) | 0.789 | 0.82 (0.44–1.51) | 0.517 |

cART, combination antiretroviral therapy; JHS, junior high school; SHS, senior high school; BMI, body mass index.

participants based on HIV status are shown in the supplementary attachment (S1–S3 Tables in S1 File).

Concerning RSP, female gender, former smoking, medium-to-high biomass exposure and being a cART-treated HIV patient were associated with increased odds in unadjusted logistic regression models. In multivariable logistic regression models, age, female gender, being unmarried, previous cigarette smoking and medium-to-high biomass exposure were associated with increased odds of restrictive lung pattern (Table 4). The associated factors of RSP among various patient groups are shown in the supplementary attachment (S4–S6 Tables in S1 File).

## Association between respiratory symptoms and abnormal spirometric patterns

In all the study participants with OSP, 14 (31.8%) reported having regular cough, 16 (36.4%) reported regular phlegm production, and 12 (16.7%) reported frequent wheezing or dyspnoea on exertion. Among the participants with RSP, those reporting regular cough or frequent wheeze or dyspnoea on exertion were 24 (16.7) while 34 (23.6%) reported regular phlegm production. In logistic regression analyses, regular cough, frequent wheeze and dyspnoea on exertion were associated with increased odds of having OSP (Fig 2). Participants with RSP had increased odds of frequent wheeze and dyspnoea on exertion in unadjusted logistic regression

**Table 3. Factors associated with an obstructive spirometric pattern from logistic regression analyses.**

| | Unadjusted model | | Adjusted model | |
|---|---|---|---|---|
| | OR (95% CI) | p | OR (95% CI) | p |
| Age (per 1 yr increase) | 0.79 (0.57–1.07) | 0.745 | 0.91 (0.73–1.41) | 0.576 |
| Females (reference = males) | 3.18 (1.24–8.85) | 0.005 | 2.46 (1.09–5.13) | 0.031 |
| Unmarried (reference = married) | 2.81 (1.35–5.83) | 0.006 | 2.31 (0.86–5.19) | 0.125 |
| Alcohol intake (reference = non-drinkers) | 0.53 (0.22–1.3) | 0.166 | 0.69 (0.19–1.489) | 0.631 |
| Smoking (Reference: never smoked) | | | | |
| Current | 1.38 (0.3–6.28) | 0.676 | 1.87 (0.36–9.75) | 0.457 |
| Former | 2.67 (1.13–4.07) | <0.002 | 1.92 (1.04–3.89) | <0.001 |
| Medium/high biomass exposure (reference = low biomass exposure) | 1.49 (1.13–3.25) | 0.004 | 3.07 (1.16–8.73) | 0.019 |
| Respiratory symptoms (reference = no respiratory symptom) | 2.35 (1.26–4.4) | 0.008 | 1.89 (1.11–5.08) | 0.029 |
| HIV status (reference: non-HIV) | | | | |
| cART-naïve | 2.24 (0.93–5.4) | 0.073 | 1.29 (0.69–3.68) | 0.548 |
| cART-treated | 3.08 (1.33–7.15) | 0.009 | 1.79 (1.03–4.18) | 0.037 |
| Employment (reference = employed) | | | | |
| Self-employed | 0.51 (0.23–1.17) | 0.112 | 0.94 (0.43–2.51) | 0.481 |
| Unemployed | 2.46 (1.15–5.23) | 0.02 | 3.26 (1.19–8.95) | 0.042 |
| Education (reference: tertiary) | | | | |
| None | 2.11 (1.08–6.71) | 0.044 | 1.72 (0.79–6.04) | 0.533 |
| Basic/JHS | 0.91 (0.43–1.9) | 0.632 | 0.77 (0.32–1.26) | 0.257 |
| SHS/Tech | 1.02 (0.36–2.84) | 0.583 | 0.63 (0.41–1.73) | 0.358 |
| BMI categories (reference = normal) | | | | |
| Underweight | 3.11 (1.27–10.21) | <0.001 | 3.01 (1.12–10.16) | 0.026 |
| Overweight | 1.09 (0.54–2.28) | 0.691 | 0.98 (0.59–2.71) | 0.864 |
| Obese | 1.12 (0.53–2.78) | 0.748 | 1.06 (0.49–4.02) | 0.877 |

cART, combination antiretroviral therapy; JHS, junior high school; SHS, senior high school; BMI, body mass index.

analysis, but there was no association between respiratory symptoms and RSP in multivariable analyses (Fig 3).

## Discussion

### Major findings

This is the first study to report the prevalence of abnormal spirometry in HIV patients in Ghana. The major findings of the study were: 1) The prevalence of OSP was higher among the higher in both cART-treated and cART-naïve HIV patient groups compared to non-HIV control, whereas the prevalence of RSP is similar among the study groups; 2) respiratory symptoms were common among HIV patients compared to non-HIV controls and were associated with abnormal spirometry; and 3) the major factors associated with OSP were female gender, former cigarette smoking, exposure to medium-to-high levels of biomass, presence of a respiratory symptom, unemployment and underweight and that of RSP were age, female gender, being unmarried, medium-to-high biomass exposure and being self-employed or unemployed.

### Abnormal spirometric patterns

The higher prevalence of OSP found in cART-treated HIV patients in our study is consistent with studies from other countries in sub-Saharan Africa. In the South African population, van

**Table 4. Factors associated with a restrictive spirometric pattern from logistic regression analyses.**

| | Unadjusted model | | Adjusted model | |
|---|---|---|---|---|
| | OR (95% CI) | p | OR (95% CI) | p |
| Age (per 1 yr increase) | 0.97 (0.93–1.01) | 0.061 | 0.97 (0.65–1.29) | 0.114 |
| Females (reference = males) | 1.67 (1.09–3.42) | 0.013 | 1.74 (1.05–4.29) | 0.041 |
| Unmarried (reference = married) | 0.75 (0.5–1.14) | 0.18 | 1.31 (0.68–3.88) | 0.741 |
| Alcohol intake (reference = non-drinkers) | 1.43 (0.88–2.310 | 0.151 | 1.68 (0.96–2.95) | 0.069 |
| Smoking (reference: never) | | | | |
| Current | 0.4 (0.09–1.78) | 0.229 | 0.42 (0.09–2.95) | 0.283 |
| Former | 3.07 (1.64–8.36) | <0.001 | 2.31 (1.27–6.77) | <0.001 |
| Medium/high biomass exposure (reference = low biomass exposure) | 2.15 (1.13–5.37) | <0.001 | 1.58 (1.06–5.37) | 0.043 |
| HIV status (reference = non-HIV controls) | | | | |
| cART-naïve | 0.89 (0.53–1.73) | 0.888 | 1.67 (0.84–3.33) | 0.144 |
| cART-treated | 1.75 (1.04–2.59) | 0.016 | 4.89 (2.15–11.11) | <0.001 |
| Employment (reference = employed) | | | | |
| Self-employed | 0.72 (0.44–1.17) | 0.182 | 0.73 (0.23–1.79) | 0.407 |
| Unemployed | 1.83 (1–3.34) | 0.05 | 2.06 (0.92–4.14) | 0.061 |
| Education (reference = tertiary) | | | | |
| None | 1.89 (0.99–3.73) | 0.067 | 2.12 (0.8–5.65) | 0.131 |
| Basic/JHS | 0.99 (0.61–1.62) | 0.979 | 0.97 (0.45–2.09) | 0.94 |
| SHS/Tech | 0.49 (0.23–1.06) | 0.069 | 0.43 (0.17–1.08) | 0.071 |
| BMI categories (reference = normal BMI) | | | | |
| Underweight | 0.84 (0.36–1.94) | 0.678 | 1.31 (0.52–3.29) | 0.562 |
| Overweight | 0.95 (0.57–1.57) | 0.837 | 0.65 (0.34–1.16) | 0.148 |
| Obese | 0.87 (0.48–1.6) | 0.661 | 0.71 (0.36–1.14) | 0.333 |

cART, combination antiretroviral therapy; JHS, junior high school; SHS, senior high school; BMI, body mass index.

Riel *et al* reported a higher prevalence of OSP in cART-treated HIV patients (34.2%) compared to cART-naïve HIV patients (11.2%). However, that study used a cut-off of 20%-LLN instead of the globally accepted 5%-LLN, which was used to characterize OSP in our study, and this may explain the usually higher prevalence of OSP reported in that particular study [18]. In

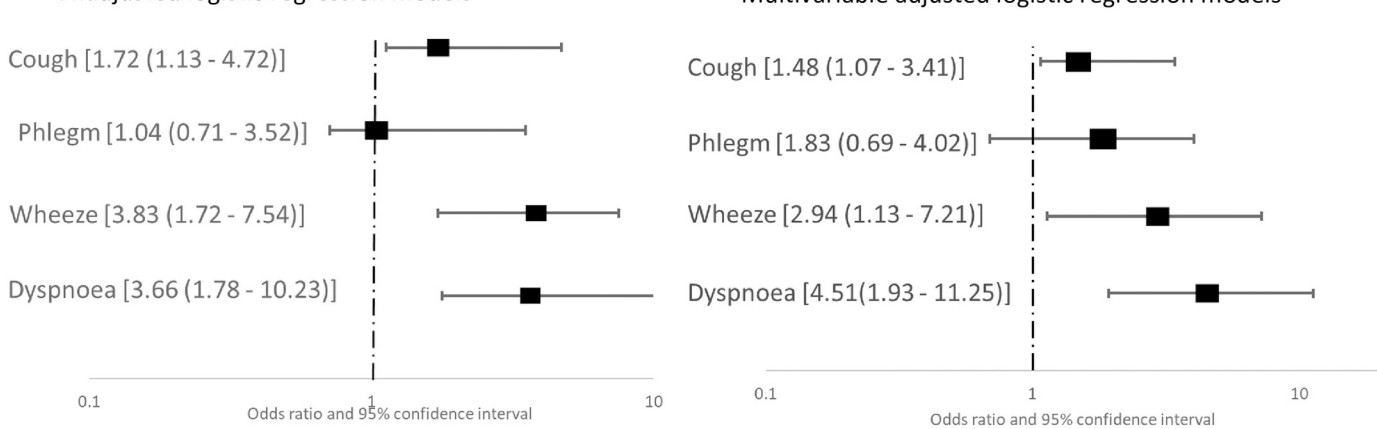

**Fig 2. Association between respiratory symptoms and obstructive spirometric pattern from unadjusted (left) and adjusted (right) logistic regression models.**

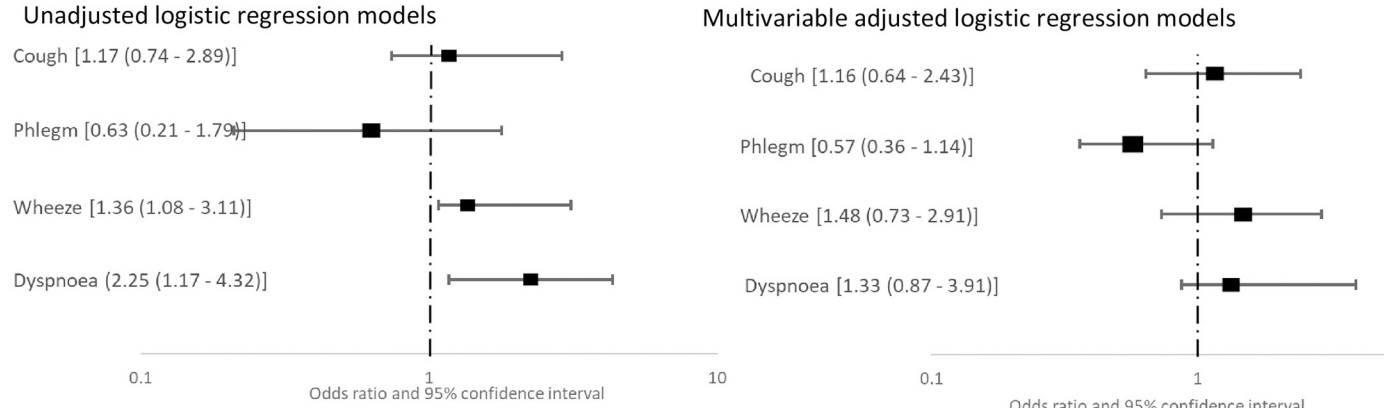

**Fig 3. Association between respiratory symptoms and restrictive spirometric pattern from unadjusted (left) and adjusted (right) logistic regression models.**

contrast to the findings of the current study, Pefura-Yone *et al* reported lower prevalences of OSP, which was similar among the HIV patients and non-HIV controls using the 5%-LLN cut-off [17]. Other cross-sectional studies in HIV patients in SSA have reported OSP prevalence between 3.1–15.4% [7, 11, 15–18], and a metanalysis reported the prevalence of OSP in HIV patients from 30 observational studies involving 151, 686 participants around the globe to be between 5.6% and 10.6% [8], similar to what was found in this study. Few studies have compared the prevalence of OSP in cART-treated HIV patients to that of cART-naïve HIV patients and non-HIV controls.

The findings of this study indicated that the prevalence of RSP was similar among cART-naïve HIV patients and non-HIV controls, but more common in cART-treated HIV patients. Most spirometry studies conducted in HIV patients and other populations in SSA do not report the prevalence and associated factors of RSP, limiting our ability to compare the prevalence of RSP found in our study to other spirometry studies in SSA. The patterns of abnormal spirometry that were reported in this study differed from what was reported by Drummond *et al* reported in HIV patients in the United States. He found that, in a total of 37% abnormal spirometry, the majority of the patients had OSP (27% for OSP, compared to 10% for RSP) [10]. The prevalence of RSP observed in our non-HIV controls (16.7%) was similar to 7–14% reported in the general US population [10], but higher than 5.9% reported in the older population [24] and 4.7% reported in Peruvian adults [25], and far lower than to 36% reported by Schultz *et al* in Brazilians [26]. The variation in the prevalence of abnormal spirometric patterns may be due to different reference equations applied in various studies [26] and the fact that various insults may affect pulmonary health across various geographical locations.

The accuracy of the definition of RSP used in this and many other studies, $FVC < 80\%$ predicted and $FEV_1/FVC > LLN$, in diagnosing RSP has been reported to be poor with low sensitivity but high negative predictive value to exclude patients without restrictive lung disease [27]. On the other hand, some studies reported that using $FVC < 80\%$ predicted alone, without $FEV_1/FVC > LLN$, improves specificity and negative prediction of diagnosing restrictive lung disease [13]. These inconsistent definitions of RSP in literature and the failure of current guidelines to address this have resulted in the underappreciation of RSP values in spirometric studies. However, the utility and importance of RSP have been highlighted in some studies as a predictor of cardiovascular morbidity and mortality. For instance, the Cardiovascular Health Study reported that RSP is a significant predictor of CVD mortality and non-CVD mortality in older Caucasians [27]. Similarly, the TESOAD study reported that, although consistent RSP

is found in 50% of patients with one-time spirometry, it is associated with an increased risk of all-cause mortality, and specifically, an increased risk of dying from diabetes and heart diseases [14]. Therefore, there is a need for further investigation into the relevance of the high burden of RSP found in HIV patients and non-HIV controls in our study to characterize them properly to prevent future CVD events and deaths.

## Factors associated with abnormal spirometry

The female gender was associated with an increased likelihood of having abnormal spirometry in this study. A systematic review has shown that females are more likely to report asthma and airway obstruction compared to men, but abnormal spirometry is similar in both genders [28, 29]. The association between female sex and abnormal spirometry found in our study may be due to the high representation of females in our study population (67.2%). This study found an association between unemployment and OSP, consistent with what has been reported in multicentre studies like BOLD [30] and PLATINO [31] studies. The debilitating nature of airway obstruction may have led to the cessation of work in those patients [30, 31]. Unlike many studies that reported an association between cigarette smoking and airway obstruction, this study did not find any association between current smoking and abnormal spirometry, possibly due to the low proportion of smokers in all our study participants (3.4%), but higher in HIV patient groups. Analysis of Demographic Health Surveys of 25 countries in Africa indicated that HIV patients with low socioeconomic status, that is the poor, less educated and manual workers, are more likely to smoke [32]. The prevalence of smoking in this study is similar to that of the general Ghanaian population (4.85%) as reported by the 2014 Ghana Demographic Health Survey [33]. On the other hand, previous smoking was associated with abnormal spirometric patterns and this is consistent with other studies [10, 26]. The findings of this study indicate that biomass exposure was associated with abnormal spirometry consistent with other reported studies [7, 15]. Biomass fumes contain about 50 air pollutants, with significant exposure similar to regular cigarette smoking [34]. These pollutants can irritate the respiratory tract by causing inflammation and tissue injury such as the excessive release of matrix metalloproteinases and increased hyaluronidase activity [35] that damage the lungs. In this study, no association was found between abnormal spirometry and cART regimen and CD4 cell count, similar to what was reported in other HIV studies [7, 10, 15].

## Respiratory symptoms

The prevalence of respiratory symptoms in our HIV population is similar to what has been reported in other sub-Saharan African populations. In urban Ugandan HIV patients, Ddungu *et al* reported the prevalence of any respiratory symptom to be 45%, with 30% reporting chronic cough, 21% reporting frequent phlegm production and wheezing and 26% reporting dyspnoea [16]. Similarly, in Cameroonian HIV patients, Pefura-Yone *et al* reported the prevalence of any respiratory symptom to be 47.5% while that of cough and dyspnoea were 25.5% and 36.4% respectively [17]. Contrary to the finding of our study, the prevalence of respiratory symptoms reported in Nigerian HIV patients was low (17%) [15] and HIV patients in South Africa had <4% of any respiratory symptoms [18]. Systematic review and meta-analysis of studies conducted in developed countries have similarly reported a high prevalence of respiratory symptoms irrespective of access to cART and high viremic control [9]. In this study, HIV status and medium-high biomass exposure were associated with respiratory symptoms in the adjusted model. Respiratory symptoms are known to be common with increasing age due to prolonged exposure to environmental toxins and decreased physiological capacity [12]. The participants in this study had a low smoking burden but high biomass exposure and

environmental pollution common in the peri-urban area might be responsible for the high prevalence of respiratory symptoms [34]. With the popularity of spirometry for screening airway obstruction, respiratory symptoms are being given less attention. However, the symptoms experienced by HIV patients indicates functional status and quality of life, even if they correlate poorly with spirometry [9].

## Limitations of the study

This study was conducted in a single primary healthcare facility in peri-urban Ghana, and hence, the findings may differ from the urban and rural populations, as well as other healthcare facilities in a different stratum. An inference of causality cannot be made from the findings of the study due to the cross-sectional design. It may be possible that some of the study participants had abnormal spirometry before HIV infection. The use of questionnaires in data collection of chronic respiratory symptoms may be affected by recall bias, which may lead to the misclassification of some study participants. This study did not assess important parameters like physical activity level, cART adherence, previous AIDS events and viral suppression which are reported to have an impact on pulmonary health. In this study, airway obstruction was defined by single spirometry. It has been reported in patients diagnosed with COPD using spirometry, 20% reversed from OSP to normal spirometry after one year of diagnosis, and this increased to 28% after two years. Among patients diagnosed with normal spirometric patterns, 11% progressed to OSP after one year and this increased to 17% after two years [36]. Therefore, categorising patients with abnormal spirometric patterns as COPD using a single spirometric measurement, as done in this study, may lead to potential misclassification. In addition, the reference equations for the Cameroonian population were employed to compute prediction values and LLN for spirometric values, and this yielded different outcomes compared to the GLI reference equations for African-Americans. The GLI equation is reported to significantly underestimate FEV1 and FVC by 9% in the Cameroonian population [22], implying significant errors in studies conducted in SSA that applied the GLI reference equation.

## Conclusion

In conclusion, this study has shown that in PLWH without any recent pulmonary infection in a peri-urban area of Ghana, the prevalence of OSP was higher among cART-treated and cART-naïve HIV patients groups compared non-HIV control. The prevalence of RSP was similar among various study groups. This study has also shed light on the various factors associated with spirometric abnormalities with the hope that future studies may investigate the possibility of using interventions targeted at these factors to reduce pulmonary dysfunction in HIV patients.

## Supporting information

**S1 File. Factors associated with abnormal spirometry in HIV patients with or without cART.**
(DOCX)

## Acknowledgments

Our sincere thanks go to the staff and patients of Atua Government Hospital who assisted in data collection, especially Mr Samuel Essel and Mrs Nneka Essel.

## Author Contributions

**Conceptualization:** Kwame Yeboah.

**Data curation:** Kwame Yeboah, Latif Musa.

**Formal analysis:** Kwame Yeboah.

**Investigation:** Latif Musa.

**Resources:** Latif Musa.

**Supervision:** Kwame Yeboah, Kweku Bedu-Addo.

**Validation:** Kweku Bedu-Addo.

**Writing – original draft:** Kwame Yeboah.

**Writing – review & editing:** Latif Musa, Kweku Bedu-Addo.

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
