## [Decision Letter · Decision Letter 0]

23 Jan 2023

PONE-D-22-21464Abnormal spirometric patterns and respiratory symptoms in HIV patients with no recent pulmonary infection in a periurban hospital in GhanaPLOS ONE

Dear Dr Kwame Yeboah,

Thank you for submitting your manuscript to PLOS ONE. After careful consideration, we feel that it has merit but does not fully meet PLOS ONE’s publication criteria as it currently stands. Therefore, we invite you to submit a revised version of the manuscript that addresses the points raised during the review process.

We look forward to receiving your revised manuscript.

Kind regards,

Zacharie Tsala Dimbuene, Ph.D.

Academic Editor

PLOS ONE

Journal Requirements:

“There was no funding for this study”

Additional Editor Comments (if provided):

Dear authors,

We have now received 3 reviewers' comments for your paper. Based on my own review and the 3 reviewers' comments, we think the paper has some potential. However, it requires some more work. Indeed, the Methods and Discussion section needs major adjustments. First, the paper is a "case-control" study; however, authors are silent about some limitations of this research design such as "selection bias". On page 6, the authors state that "the sample size was based on a prevalence of COPD in a rural Ugandan population of 16.2%" without any justification and possible implications for the current study. This study was conducted in a peri-urban area. Second, Table 2 on page 7 clearly shows that participants are not alike on many characteristics. What are the implications on findings and conclusion of the study. Third, authors are using almost interchangeably the terms "determinants" and "associated factors" along the manuscript. I wonder if the are aware of the conceptual difference between them.

Reviewers' comments:

Reviewer's Responses to Questions

**Comments to the Author**

1. Is the manuscript technically sound, and do the data support the conclusions?

Reviewer #1: Partly

Reviewer #2: Yes

Reviewer #3: Partly

2. Has the statistical analysis been performed appropriately and rigorously? 

Reviewer #1: No

Reviewer #2: Yes

Reviewer #3: Yes

3. Have the authors made all data underlying the findings in their manuscript fully available?

Reviewer #1: No

Reviewer #2: Yes

Reviewer #3: No

4. Is the manuscript presented in an intelligible fashion and written in standard English?

Reviewer #1: No

Reviewer #2: Yes

Reviewer #3: Yes

5. Review Comments to the Author

Reviewer #1: The authors address the issue of a comorbidity that often goes unnoticed.

1. The introduction is well developed.

2. In the M&M section there are some aspects that should be taken into account:

- Define better the inclusion and exclusion criteria of this proposal

- Use (or not) of bronchodilators, previous history of tuberculosis, previous history of AIDS events, etc.

- Please define former smoker. Please, could you provide information on the number of pack-year?

- In PLWH on ART, how many had an undetectable viral load (define)? How long have they been on ART?

- Could you provide information about physical activity? It has been implicated on lung health.

- Please assess FEV1/FVC ratio 80%.

- Prior to multivariate analysis it would be advisable to do a collinearity analysis.

3. The results and discussion section could be conditioned by the above questions.

Reviewer #2: Reviewer’s Comments

Abstract

The authors should state what odd ratios and the p values were with regards to the major determinants of OSP and RSP.

Sample Size

Why was a prevalence study in Uganda used for the calculation of the sample size?

Results

Table 1

Past smoking looks like a predisposing factor for the abnormalities seen in the spirometry when the HIV patients are compared to the non-HIV patients. This looks like the major factor to account for the results that was gotten. How did the authors control for this since it seems to be a major confounding factor?

Determinants of abnormal spirometry patterns

The authors found the female gender, unemployment, underweight to be associated with increased odds of having OSP. There were no explanations for these findings in the results section.

The authors also found female gender, being unmarried, being self-employed or unemployed to be associated with increased odds of restrictive lung pattern. However, there are no explanations for these findings in their results. Can they explain the association between these variables and the spirometry abnormalities.

Reviewer #3: 1. Study title: is satisfactory

2. Abstract: is satisfactory but change the sentence : the Spirometry is an accurate method of diagnosing pulmonary dysfunction n people living with HIV (PLWH).

Spirometry is a supplementary test that aids in the diagnosis a pulmonary disease

Introduction : What is the rational for this study and clinical importance of the study?

After the hypothesis , please restate the objective of your study

Methodology

The definition of biomass use/exposure and sentence has no reference, the author should include a reference(s) -Medium-high biomass exposure was defined as those who reported using firewood, sawdust, cow-dung or corn cubs as a fuel source for work or cooking at home on most days for more than 6 months

The author did not mention how the modified version of the American Thoracic Society respiratory questionnaire was administered, and whether it was in English or Local dialect, If in local dialect was the questionnaire validated.

what is reason for not collecting CD4 Count in the non HIV controls?

Is there any reason for using lower limit of normal than the fixed ratio of FEV/FVC

Sample size calculation lacks clarity, please include the sample size calculation with relevant reference. How did you arrived at the selected figures for the cases and controls

Results

The author needs to tell the reader how many where approached in both the cases and control , the attrition rate in each group and the reason for not completing the study. How many patients and controls had poor spirometry results and was there problem encountered during questionnaire administration?

Table 1 - the unit of the parameter should be in closed bracket()

Table 2- There are 2 respiratory symptoms parameters with different results, the author should verify these results

Discussion

It is not a robust discussion, there were not enough explanation for incongruent result when making comparison.

This sentence was not fully discussed and interpreted : Drummond et al reported in HIV patients in the United States that, out of 37% of HIV patients who had abnormal spirometry, 10% had RSP while 27% had OSP [8], a pattern contrary to what was observed in our study population with RSP higher than OSP.

What is the reason for higher number of smoker in HIV? I would suggest you look for previous work on smoking in HIV in Sub-Saharan Africa to support your discussion.

Regarding the respiratory symptoms , the Nigerian study you cited had an objective and design that were different from this study . I would suggest similar study below in Nigeria with almost similar methods to your study to support your discussion- pls note it is at your discretion

Onyedum, C. C., Chukwuka, J. C., Onwubere, B. J., Ulasi, I. I., & Onwuekwe, I. O. (2010). Respiratory symptoms and ventilatory function tests in Nigerians with HIV infection. African health sciences, 10(2), 130–137.

The High prevalence respiratory symptom in HIV , Is there a relation between gender and ,high biomass exposure

There discussion regarding these association should be more vibrant

It has been that about 20% and 28% of patients initially diagnosed with OSP reversed to normal

pattern after one and two years, respectively. Pls reword this sentence for clarity and adduce reason for the reversal

.if the The GLI equation is reported to significantly underestimate FEV1 and FVC by 9% in the Cameroonian population [32], why not use the Cameroun study or other reference equation in West Africa rather the GLI?

6. PLOS authors have the option to publish the peer review history of their article (what does this mean?). If published, this will include your full peer review and any attached files.

Reviewer #1: No

Reviewer #2: **Yes: **Christian Obirikorang

Reviewer #3: **Yes: **Olufemi Desalu

---

## [Author Response · Author response to Decision Letter 0]

13 Mar 2023

5. Review Comments to the Author

Reviewer #1: The authors address the issue of a comorbidity that often goes unnoticed.

1. The introduction is well developed.

2. In the M&M section there are some aspects that should be taken into account:

- Define better the inclusion and exclusion criteria of this proposal- Use (or not) of bronchodilators, previous history of tuberculosis, previous history of AIDS events, etc.

Response: We have included the eligibility criteria in the methodology section.

- Please define former smoker. Please, could you provide information on the number of pack-year?

Response: Most of our former smokers were occasional smokers who used to smoke during ceremonies or with friends. Most of them found it difficult to recollect the estimated frequency and number of sticks they smoked previously. That is the reason we chose to analyze only current smoking to avoid high levels of recall bias.

- In PLWH on ART, how many had an undetectable viral load (define)? How long have they been on ART?

Response: Due to logistical constraints at the time of data collection, viral load measurement in southern Ghana was temporarily suspended and when they resume measurement, there was a long queue of samples to be analyzed. This hampered our ability to measure viral load. We have therefore acknowledged this in the limitation section.

- Could you provide information about physical activity? It has been implicated on lung health.

Response: We did not collect any data on physical activity levels. We have acknowledged this in our limitation section.

- Please assess FEV1/FVC ratio 80%.

Response: We have provided assessments of abnormal spirometric measurements based on the fixed ratio in Table 1. 

- Prior to multivariate analysis it would be advisable to do a collinearity analysis.

Response: We performed collinearity analysis and in all cases, valence inflation factor was between 1.07 – 1.51 and tolerance was 0.71 – 0.96. Also, we excluded CD4 cell counts, HIV duration and cART drugs due to multicollinearity with other factors in the models.

3. The results and discussion section could be conditioned by the above questions.

Reviewer #2: Reviewer’s Comments

Abstract

The authors should state what odd ratios and the p values were with regards to the major determinants of OSP and RSP.

Response: We have kindly provided the odds ratios and p-values in the abstract section.

Sample Size

Why was a prevalence study in Uganda used for the calculation of the sample size?

Response: We thank the reviewer for drawing our attention to this. We have corrected the sample size calculation to include two studies, one in HIV patients and the other in non-HIV population, conducted in rural Ugandan population. We believe that since lung abnormalities are related to urbanization, the rural setting of the Uganda studies closely resemble our study setting. Hence, that informed us to use the data from rural Uganda.

Results

Table 1

Past smoking looks like a predisposing factor for the abnormalities seen in the spirometry when the HIV patients are compared to the non-HIV patients. This looks like the major factor to account for the results that was gotten. How did the authors control for this since it seems to be a major confounding factor?

Response: We have kindly included previous smoking in our revised analysis.

Determinants of abnormal spirometry patterns

The authors found the female gender, unemployment, underweight to be associated with increased odds of having OSP. There were no explanations for these findings in the results section.

The authors also found female gender, being unmarried, being self-employed or unemployed to be associated with increased odds of restrictive lung pattern. However, there are no explanations for these findings in their results. Can they explain the association between these variables and the spirometry abnormalities.

Response: We have kindly expanded the discussion section to include these determinants of abnormal spirometric patterns in the “Determinants of abnormal spirometric patterns” section of the discussion.

Reviewer #3: 1. Study title: is satisfactory

2. Abstract: is satisfactory but change the sentence : the Spirometry is an accurate method of diagnosing pulmonary dysfunction n people living with HIV (PLWH).

Spirometry is a supplementary test that aids in the diagnosis a pulmonary disease

Response: We have kindly revised the abstract according to the recommendations of the reviewers.

Introduction : What is the rational for this study and clinical importance of the study?

After the hypothesis , please restate the objective of your study

Response: We hae restated the objective after the rationale of the study.

Methodology

The definition of biomass use/exposure and sentence has no reference, the author should include a reference(s) -Medium-high biomass exposure was defined as those who reported using firewood, sawdust, cow-dung or corn cubs as a fuel source for work or cooking at home on most days 

Response: We have kindly included reference to that statement and adjusted the statement to reflect that reference.

The author did not mention how the modified version of the American Thoracic Society respiratory questionnaire was administered, and whether it was in English or Local dialect, If in local dialect was the questionnaire validated.

Response: For the majority of the participants, they could comprehend basic English and hence, the questionnaire was administered in English. However, there were few participants with limited English conprehension that we translated the questionnaire into their local dialect before administration. The questionnaire, unfortunately, had not been validated. However, the symptoms being asked from the questionnaire are similar to what we normally ask the patients during routine clinical visits and hence, we expected them to understand.

what is reason for not collecting CD4 Count in the non HIV controls?

Response: We could not measure CD4 lymphocyte counts in non-HIV participants due to logistical constraints. There was a lot of pressure on the only device to assay CD4 cell count at the region where the study was conducted, with large quantities of samples and fewer reagents available. Therefore, we could only lobby the needed authorities for the assaying of only registered as HIV positive.

Is there any reason for using lower limit of normal than the fixed ratio of FEV/FVC

Response: We have presented both the results for the lower limit of normal and fixed ratios in the revise manuscript. 

Sample size calculation lacks clarity, please include the sample size calculation with relevant reference. How did you arrived at the selected figures for the cases and controls

Response: We have kindly provided further information of the sample size calculation.

Results

The author needs to tell the reader how many where approached in both the cases and control , the attrition rate in each group and the reason for not completing the study. How many patients and controls had poor spirometry results and was there problem encountered during questionnaire administration?

Response: we have kindly provided information on the number of participants initially recruited and those who were not able to produce the required manoeuvre for accurate spirometric recordings.

Table 1 - the unit of the parameter should be in closed bracket

Response: We kindly prefer to use commas for units of continuous variables and counts (percentages) for categorical variables.

Table 2- There are 2 respiratory symptoms parameters with different results, the author should verify these results.

Response: We have humbly removed the wrong data from the table.

Discussion

It is not a robust discussion, there were not enough explanation for incongruent result when making comparison. This sentence was not fully discussed and interpreted : Drummond et al reported in HIV patients in the United States that, out of 37% of HIV patients who had abnormal spirometry, 10% had RSP while 27% had OSP [8], a pattern contrary to what was observed in our study population with RSP higher than OSP.

Response: We have kindly rephrased the sentence to make it clearer and we have explained the sentence.

What is the reason for higher number of smoker in HIV? I would suggest you look for previous work on smoking in HIV in Sub-Saharan Africa to support your discussion.

Response: We have explained that the HIV patients who smoked were more likely to be from low socioeconomic status and we have also included participants who were former smokers in the multivariable analysis.

Regarding the respiratory symptoms , the Nigerian study you cited had an objective and design that were different from this study . I would suggest similar study below in Nigeria with almost similar methods to your study to support your discussion- pls note it is at your discretion

Onyedum, C. C., Chukwuka, J. C., Onwubere, B. J., Ulasi, I. I., & Onwuekwe, I. O. (2010). Respiratory symptoms and ventilatory function tests in Nigerians with HIV infection. African health sciences, 10(2), 130–137.

response: we appreciate the recommendation by the reviewer and have done so

The High prevalence respiratory symptom in HIV , Is there a relation between gender and, high biomass exposure?

Response: We have kindly modified that statement in the discussion section to reflect what is in the revised results. Only HIV status and high biomass exposure were associated with respiratory symptoms.

There discussion regarding these association should be more vibrant

It has been that about 20% and 28% of patients initially diagnosed with OSP reversed to normal pattern after one and two years, respectively. Pls reword this sentence for clarity and adduce reason for the reversal

Response: We have kindly rephrased the sentence to make it clearer and we have explained the sentence. The main reason is that one-time spirometry may misclassify patients and hence, cannot be fully relied on to diagnose COPD.

if the The GLI equation is reported to significantly underestimate FEV1 and FVC by 9% in the Cameroonian population [32], why not use the Cameroun study or other reference equation in West Africa rather the GLI?

Response: We have re-analyzed the data using the reference equation from the Cameroonian study to determine the lower limit of normal and hence abnormal spirometric patterns.

---

## [Decision Letter · Decision Letter 1]

15 May 2023

PONE-D-22-21464R1Abnormal spirometric patterns and respiratory symptoms in HIV patients with no recent pulmonary infection in a periurban hospital in GhanaPLOS ONE

Dear Dr. Yeboah,

Thank you for submitting your manuscript to PLOS ONE. After careful consideration, we feel that it has merit but does not fully meet PLOS ONE’s publication criteria as it currently stands. Therefore, we invite you to submit a revised version of the manuscript that addresses the points raised during the review process.

I commend that the manuscript has improved but there still are many issues to address. Attached my comments and suggestions.

We look forward to receiving your revised manuscript.

Kind regards,

Zacharie Tsala Dimbuene, Ph.D.

Academic Editor

PLOS ONE

Reviewers' comments:

Reviewer's Responses to Questions

**Comments to the Author**

1. If the authors have adequately addressed your comments raised in a previous round of review and you feel that this manuscript is now acceptable for publication, you may indicate that here to bypass the “Comments to the Author” section, enter your conflict of interest statement in the “Confidential to Editor” section, and submit your "Accept" recommendation.

Reviewer #2: All comments have been addressed

2. Is the manuscript technically sound, and do the data support the conclusions?

Reviewer #2: Yes

3. Has the statistical analysis been performed appropriately and rigorously? 

Reviewer #2: Yes

4. Have the authors made all data underlying the findings in their manuscript fully available?

Reviewer #2: Yes

5. Is the manuscript presented in an intelligible fashion and written in standard English?

Reviewer #2: Yes

6. Review Comments to the Author

Reviewer #2: All major comments raised has been resolved. There are no more issues with the manuscript as it stands now.

7. PLOS authors have the option to publish the peer review history of their article (what does this mean?). If published, this will include your full peer review and any attached files.

Reviewer #2: **Yes: **Christian Obirikorang

---

## [Author Response · Author response to Decision Letter 1]

10 Aug 2023

Comment

This version has substantively improved. However, there still is window for improvements to reach the standards of PLOS One. I have extensively edited and annotated the manuscript. I urge the authors to pay attention on my comments in the manuscript. Here, I provide a brief on those comments.

For the entire manuscript, checks for compliance the use of “We” and “I” 

Response: We have kindly rephrased all the sentences with the pronoun “we” 

Abstract. See manuscript on a few suggested changes 

Response: We have kindly effected all the suggested changes in the abstract.

Introduction. Some affirmations lack references 

Response: We have kindly added references to the affirmations queried.

Methods. The logistic regression is misphrased. Authors are expected to rephrase the formulation of logistic regression (see p. 7) 

Response: We have revised the statement describing the utility of logistic regression as suggested.

Results. Tables 2—4: All categorical variables need a reference category (RC). Females and Unmarried are categorical variables as well

Response: We have added reference categories to the categorical variables in the Tables 2 – 4.

Please use “Factors associated” but “determinants” which has a “causation” connotation 

Response: The term “determinants” has been changed to “factors associated with” as suggested.

---

## [Decision Letter · Decision Letter 2]

23 Apr 2024

PONE-D-22-21464R2Abnormal spirometric patterns and respiratory symptoms in HIV patients with no recent pulmonary infection in a periurban hospital in GhanaPLOS ONE

Dear Dr. Yeboah,

Thank you for submitting your manuscript to PLOS ONE. After careful consideration, we feel that it has merit but does not fully meet PLOS ONE’s publication criteria as it currently stands. Therefore, we invite you to submit a revised version of the manuscript that addresses the points raised during the review process.

**ACADEMIC EDITOR: Kindly address Reviewer's comments.**

We look forward to receiving your revised manuscript.

Kind regards,

Zacharie Tsala Dimbuene, Ph.D.

Academic Editor

PLOS ONE

Journal Requirements:

Reviewers' comments:

Reviewer's Responses to Questions

**Comments to the Author**

1. If the authors have adequately addressed your comments raised in a previous round of review and you feel that this manuscript is now acceptable for publication, you may indicate that here to bypass the “Comments to the Author” section, enter your conflict of interest statement in the “Confidential to Editor” section, and submit your "Accept" recommendation.

Reviewer #4: (No Response)

Reviewer #5: All comments have been addressed

2. Is the manuscript technically sound, and do the data support the conclusions?

Reviewer #4: Partly

Reviewer #5: Yes

3. Has the statistical analysis been performed appropriately and rigorously? 

Reviewer #4: Yes

Reviewer #5: Yes

4. Have the authors made all data underlying the findings in their manuscript fully available?

Reviewer #4: Yes

Reviewer #5: Yes

5. Is the manuscript presented in an intelligible fashion and written in standard English?

Reviewer #4: Yes

Reviewer #5: Yes

6. Review Comments to the Author

Reviewer #4: Manuscript under review:

Abnormal spirometric patterns and respiratory symptoms in HIV patients with no recent pulmonary infection in a periurban hospital in Ghana

Summary of Research:

This manuscript does an excellent job establishing for the first time in Ghana, prevalence of spirometric abnormalities amongst people living with HIV (PLWH) considering their treatment statuses. It further revealed the presence of both obstructive and restrictive spirometric abnormalities with the former showing great significance amongst PLWH compared to the general populace. This significant difference of obstructive spirometric patterns found between PLWH and the non-HIV control was revealed by only one of two methods (i.e. the lower limits of normal method and the fixed ratio method) employed by the authors. Yeboah et al. should comment on the demonstrated difference in sensitivity revealed by these two reference methods. Yeboah et al. could not indicate in their results, the statistical difference in prevalence of obstructive spirometric pattern between treated and naïve-HIV patients but however, made a submission on that in their conclusion. The authors in addition, took interest in investigating and reporting various factors that had associations with the prevailing spirometric abnormalities. The study highlights the clinical relevance of spirometry and hence, the need to look beyond the over-emphasized opportunistic infection as the cause of most respiratory symptoms in PLWH. The study however, did not comment or report on any findings concerning mixed obstructive and restrictive spirometric patterns as this is a possible finding in spirometry. All spirometric patterns’ definition specified by the authors excluded definition for mixed obstructive and restrictive pattern spirometric abnormality. I suggest the authors include the definition for mixed spirometric findings and as well report on their findings as this is relevant to the current study. This reasearch work will serve as a valuable reference for all future spirometric investigations amongst people living with HIV.

Areas for Improvements:

There are no major areas for improvement. There are however, a few minor issues the authors need to tackle. These are specified below.

1) Under “results” in “abstract” (page 2), line 9; review punctuation mark before the word “and”.

2) Under “results” in “abstract” (page 2), line 10; replace the punctuation mark before the word “medium” with the word –“and”.

3) Under “study participants” in “methods” (page 4), line 6; review the location stated for Atua Government Hospital.

4) Under “general characteristics of study participants” in “results” (page 7), line 4; review the phrase – “were likely to be” since you already found and know the result.

5) Under “general characteristics of study participants” in “results” (page 7), line 7; review the phrase – “were likely to be” since you already found and know the result.

6) Under “general characteristics of study participants” in “results” (page 7), line 7; indicate the proportion allocated to the underweights.

7) Under “general characteristics of study participants” in “results” (page 9), Table 1; indicate with Greek symbols, groups within which significant differences exist in the “obstructive” results under “abnormal spirometry by LLN, n (%)”

8) Under “general characteristics of study participants” in “results” (page 9), Table 1; indicate with Greek symbols, groups within which significant differences exist in results for “Respiratory symptoms, n (%)”

9) Under “Respiratory symptoms” in “results” (page 9), Fig 1; indicate where in the document this “fig 1” could be located as actual figure is omitted.

10) Under “Association between respiratory symptoms and abnormal spirometric patterns” in “results” (page 13), Fig 2; indicate where in the document this “Fig 2” could be located as actual figure is omitted.

11) Under “Association between respiratory symptoms and abnormal spirometric patterns” in “results” (page 13), Fig 3; indicate where in the document this “Fig 3” could be located as actual figure is omitted.

12) Under “Abnormal spirometric patterns” in “discussion” (page 14), line 1; do you intend to say higher or high and in reference to which of the three groups of study?

13) Under “Limitations of study” (page 18), line 17; correct the word- “ad” before the wors –“this”.

14) Under “Conclusion” in “discussion” (page 18), line 2;- use the superlative of the word –“high”.

Other points:

There are no other points to consider. I will however be interested to look at a revised version of this manuscript.

Reviewer #5: (No Response)

7. PLOS authors have the option to publish the peer review history of their article (what does this mean?). If published, this will include your full peer review and any attached files.

Reviewer #4: **Yes: **Samuel Essel

Reviewer #5: No

---

## [Author Response · Author response to Decision Letter 2]

6 May 2024

Response: We have kindly gone through all the references and they are current and appropriate. None had been retracted.

1. Areas for Improvements:

There are no major areas for improvement. There are however, a few minor issues the authors need to tackle. These are specified below.

1) Under “results” in “abstract” (page 2), line 9; review punctuation mark before the word “and”.

Response: We have kindly replaced the punctuation (,) with a (;) to make the two phrases distinct.

2) Under “results” in “abstract” (page 2), line 10; replace the punctuation mark before the word “medium” with the word –“and”.

Response: We have kindly corrected the punctuation as suggested.

3) Under “study participants” in “methods” (page 4), line 6; review the location stated for Atua Government Hospital.

Response: We have kindly corrected the location with the formal name of the municipality.

4) Under “general characteristics of study participants” in “results” (page 7), line 4; review the phrase – “were likely to be” since you already found and know the result.

Response: We have kindly replaced the phrase “were likely to be” with “were mostly”.

5) Under “general characteristics of study participants” in “results” (page 7), line 7; review the phrase – “were likely to be” since you already found and know the result.

Response: We have kindly replaced the phrase “were likely to be” with “were mostly”.

6) Under “general characteristics of study participants” in “results” (page 7), line 7; indicate the proportion allocated to the underweights.

Response: We have kindly indicated the proportion of underweights in the study to be 9.1%

7) Under “general characteristics of study participants” in “results” (page 9), Table 1; indicate with Greek symbols, groups within which significant differences exist in the “obstructive” results under “abnormal spirometry by LLN, n (%)”

Response: We have kindly used asterisks to indicate group difference in abnormal spirometry as were done for other parameters.

8) Under “general characteristics of study participants” in “results” (page 9), Table 1; indicate with Greek symbols, groups within which significant differences exist in results for “Respiratory symptoms, n (%)”

Response: We have kindly used asterisks to indicate group difference in abnormal spirometry as were done for other parameters.

9) Under “Respiratory symptoms” in “results” (page 9), Fig 1; indicate where in the document this “fig 1” could be located as actual figure is omitted.

Response: Base on PLOS One submission guidelines, Figures are not supposed to be added to the main manuscript, but uploaded separately.

10) Under “Association between respiratory symptoms and abnormal spirometric patterns” in “results” (page 13), Fig 2; indicate where in the document this “Fig 2” could be located as actual figure is omitted.

Response: Base on PLOS One submission guidelines, Figures are not supposed to be added to the main manuscript, but uploaded separately.

11) Under “Association between respiratory symptoms and abnormal spirometric patterns” in “results” (page 13), Fig 3; indicate where in the document this “Fig 3” could be located as actual figure is omitted.

Response: Base on PLOS One submission guidelines, Figures are not supposed to be added to the main manuscript, but uploaded separately.

12) Under “Abnormal spirometric patterns” in “discussion” (page 14), line 1; do you intend to say higher or high and in reference to which of the three groups of study?

Response: We have kindly changed the word to higher.

13) Under “Limitations of study” (page 18), line 17; correct the word- “ad” before the wors –“this”.

Response: We have kindly corrected the word “ad” to “and”.

14) Under “Conclusion” in “discussion” (page 18), line 2;- use the superlative of the word –“high”.

Response: We have kindly changed the word to higher to correspond to the results of the study.

---

## [Editor Report · Decision Letter 3]

3 Jun 2024

PONE-D-22-21464R3Abnormal spirometric patterns and respiratory symptoms in HIV patients with no recent pulmonary infection in a periurban hospital in GhanaPLOS ONE

Dear Dr. Yeboah,

Thank you for submitting your manuscript to PLOS ONE. After careful consideration, we feel that it has merit but does not fully meet PLOS ONE’s publication criteria as it currently stands. Therefore, we invite you to submit a revised version of the manuscript that addresses the points raised during the review process.

Kindly incorporate comments from Reviewer #4  

We look forward to receiving your revised manuscript.

Kind regards,

Zacharie Tsala Dimbuene, Ph.D.

Academic Editor

PLOS ONE
---

## [Author Response · Author response to Decision Letter 3]

22 Jul 2024

I have kindly incorporated the comments of the Reviewer #4 as follows:

1. Areas for Improvements:

There are no major areas for improvement. There are however, a few minor issues the authors need to tackle. These are specified below.

1) Under “results” in “abstract” (page 2), line 9; review punctuation mark before the word “and”.

Response: We have kindly replaced the punctuation (,) with a (;) to make the two phrases distinct.

2) Under “results” in “abstract” (page 2), line 10; replace the punctuation mark before the word “medium” with the word –“and”.

Response: We have kindly replaced the punctuation mark before the word “medium” with the word “and” as suggested.

3) Under “study participants” in “methods” (page 4), line 6; review the location stated for Atua Government Hospital.

Response: We have kindly corrected the location with the formal name as the the Lower Manya Krobo Municipality.

4) Under “general characteristics of study participants” in “results” (page 7), line 4; review the phrase – “were likely to be” since you already found and know the result.

Response: We have kindly replaced the phrase “were likely to be” with “were mostly”.

5) Under “general characteristics of study participants” in “results” (page 7), line 7; review the phrase – “were likely to be” since you already found and know the result.

Response: We have kindly replaced the phrase “were likely to be” with “were mostly”.

6) Under “general characteristics of study participants” in “results” (page 7), line 7; indicate the proportion allocated to the underweights.

Response: We have kindly indicated the proportion of underweights in the study to be 9.1%

7) Under “general characteristics of study participants” in “results” (page 9), Table 1; indicate with Greek symbols, groups within which significant differences exist in the “obstructive” results under “abnormal spirometry by LLN, n (%)”

Response: We have kindly used asterisks to indicate group difference in abnormal spirometry as were done for other parameters.

8) Under “general characteristics of study participants” in “results” (page 9), Table 1; indicate with Greek symbols, groups within which significant differences exist in results for “Respiratory symptoms, n (%)”

Response: We have kindly used asterisks to indicate group difference in abnormal spirometry as were done for other parameters.

9) Under “Respiratory symptoms” in “results” (page 9), Fig 1; indicate where in the document this “fig 1” could be located as actual figure is omitted.

Response: We have kindly indicated the location in the manuscript that Fig 1 should be inserted at Page 9 line 22. Also, Fig 1 was uploaded separately as a file document as recommended by PLOS One submission guidelines.

10) Under “Association between respiratory symptoms and abnormal spirometric patterns” in “results” (page 13), Fig 2; indicate where in the document this “Fig 2” could be located as actual figure is omitted.

Response: We have kindly indicated the location in the manuscript that Fig 2 should be inserted at Page 14 line 14. Also, Fig 2 was uploaded separately as a file document as recommended by PLOS One submission guidelines.

11) Under “Association between respiratory symptoms and abnormal spirometric patterns” in “results” (page 13), Fig 3; indicate where in the document this “Fig 3” could be located as actual figure is omitted.

Response: We have kindly indicated the location in the manuscript that Fig 3 should be inserted at Page 14 line 16. Also, Fig 3 was uploaded separately as a file document as recommended by PLOS One submission guidelines.

12) Under “Abnormal spirometric patterns” in “discussion” (page 14), line 1; do you intend to say higher or high and in reference to which of the three groups of study?

Response: We have kindly revised the statement as “the prevalence of OSP was higher in both cART-treated and cART-naïve HIV patient groups compared to non-HIV control”.

13) Under “Limitations of study” (page 18), line 17; correct the word- “ad” before the wors –“this”.

Response: We have kindly corrected the word “ad” to “and”.

14) Under “Conclusion” in “discussion” (page 18), line 2;- use the superlative of the word –“high”.

Response: We have kindly revised the statement as “the prevalence of OSP was higher in both cART-treated and cART-naïve HIV patient groups compared to non-HIV control”.

---

## [Editor Report · Decision Letter 4]

29 Jul 2024

PONE-D-22-21464R4Abnormal spirometric patterns and respiratory symptoms in HIV patients with no recent pulmonary infection in a periurban hospital in GhanaPLOS ONE

Dear Dr. Yeboah,

Thank you for submitting your manuscript to PLOS ONE. After careful consideration, we feel that it has merit but does not fully meet PLOS ONE’s publication criteria as it currently stands. Therefore, we invite you to submit a revised version of the manuscript that addresses the points raised during the review process.

**ACADEMIC EDITOR: **The manuscript has significantly improved. I have now extensively edited the revised version but still, some minor changes need to be done before formal acceptance. See attached the annotated version and incorporate all suggested changes, unless otherwise. 

We look forward to receiving your revised manuscript.

Kind regards,

Zacharie Tsala Dimbuene, Ph.D.

Academic Editor

PLOS ONE
---

## [Author Response · Author response to Decision Letter 4]

14 Sep 2024

Response to editor queries:

1. Harmonize all over the places if you want spaces before and after the sign =; <; etc

Response: We have added spaces before and after all signs as suggested.

2. This is very unclear

Response: We have clarified the statement to mean: “However, the prevalence of RSP was similar among HIV patients and non-HIV controls.”

3. COPD; previous studies suggested

Response: We have kindly changed the statement as suggested.

4. This para addresses inclusion/exclusion criteria. It should have been clearer a sub-heading "Inclusion/Exclusion" and simply list/explain these criteria.

Response: We have kindly created a subheading for inclusion and exclusion criteria

5. Table 1 should have been announced earlier before aligning findings

Response: We have mentioned the Table 1 at the beginning of the paragraph as suggested.

6. Not sure the way this Table was built align with PLoS One requirements

Response: We have ensured that the Table 1 aligns with PLOSone requirements

7. all "p" in Italics please

Response: All “p” are in italics as suggested.

8. be consistent: use being female or female gender

Response: We have kindly changed all females to female gender.

9. why different font?

Response: We have harmonized all fonts in the manuscript please.

10. Is this part of main findings or not?

Response: Yes, it is part of the main findings discussing the prevalence of abnormal spirometry, which includes obstructive/restrictive patterns.

11. References: Use same font as in the manuscript

Response: The font of the references have been changed to be the same as the manuscript.

---

## [Editor Report · Decision Letter 5]

24 Sep 2024

Abnormal spirometric patterns and respiratory symptoms in HIV patients with no recent pulmonary infection in a periurban hospital in Ghana

PONE-D-22-21464R5

Dear Dr. Kwame Yeboah,

We’re pleased to inform you that your manuscript has been judged scientifically suitable for publication and will be formally accepted for publication once it meets all outstanding technical requirements.

Kind regards,

Zacharie Tsala Dimbuene, Ph.D.

Academic Editor

PLOS ONE
---

## [Editor Report · Acceptance letter]

27 Sep 2024

PONE-D-22-21464R5 

PLOS ONE

Dear Dr. Yeboah, 

I'm pleased to inform you that your manuscript has been deemed suitable for publication in PLOS ONE. Congratulations! Your manuscript is now being handed over to our production team.

Kind regards, 

on behalf of

Prof. Zacharie Tsala Dimbuene 

Academic Editor

PLOS ONE